# Coexistence or conflict: Black bear habitat use along an urban-wildland gradient

**Joanna Klees van Bommel**[1]*, **Catherine Sun**[1], **Adam T. Ford**[2], **Melissa Todd**[3], **A. Cole Burton**[1,4]

1 Department of Forest Resources Management, University of British Columbia, Vancouver, British Columbia, Canada, 2 Department of Biology, University of British Columbia Okanagan, Kelowna, British Columbia, Canada, 3 British Columbia Ministry of Forests, Coast Area Research Section, Nanaimo, British Columbia, Canada, 4 Biodiversity Research Centre, University of British Columbia, Vancouver, British Columbia, Canada

* kvb.joanna@gmail.com

**Data Availability Statement:** The data are available from the Dryad database (https://doi.org/10.5061/dryad.nvx0k6dvf).

## Abstract

The urban-wildland interface is expanding and increasing the risk of human-wildlife conflict. Some wildlife species adapt to or avoid living near people, while others select for anthropogenic resources and are thus more prone to conflict. To promote human-wildlife coexistence, wildlife and land managers need to understand how conflict relates to habitat and resource use in the urban-wildland interface. We investigated black bear (*Ursus americanus*) habitat use across a gradient of human disturbance in a North American hotspot of human-black bear conflict. We used camera traps to monitor bear activity from July 2018 to July 2019, and compared bear habitat use to environmental and anthropogenic variables and spatiotemporal probabilities of conflict. Bears predominantly used areas of high vegetation productivity and increased their nocturnality near people. Still, bears used more high-conflict areas in summer and autumn, specifically rural lands with ripe crops. Our results suggest that bears are generally modifying their behaviours in the urban-wildland interface through spatial and temporal avoidance of humans, which may facilitate coexistence. However, conflict still occurs, especially in autumn when hyperphagia and peak crop availability attract bears to abundant rural food resources. To improve conflict mitigation practices, we recommend targeting seasonal rural attractants with pre-emptive fruit picking, bear-proof compost containment, and other forms of behavioural deterrence. By combining camera-trap monitoring of a large carnivore along an anthropogenic gradient with conflict mapping, we provide a framework for evidence-based improvements in human-wildlife coexistence.

## Introduction

The urban-wildland interface, where human development borders natural habitat, is increasing with an expanding human footprint [1, 2]. This has important consequences for wildlife; some animals are displaced by habitat loss due to direct and adjacent human development, while others are able to adjust their behaviours to use human areas [3]. For instance, some species have been found to reduce movement [4] and increase nocturnal activity [5] when using

**Funding:** Funding was received from a National Geographic Early Career Grant (EC-336R-18; https://www.nationalgeographic.org/society/grants-and-investments/) awarded to JKvB, the University of British Columbia's Faculty of Forestry (no grant number; https://forestry.ubc.ca/), and the Natural Sciences and Engineering Research Council of Canada (NSERC; https://www.nserc-crsng.gc.ca/index_eng.asp), through a Canada Graduate Scholarship to JKvB (no grant number) and NSERC Discovery Grant (DGECR-2018-00413) to ACB. The funders had no role in study design, data collection and analysis, decision to publish, or preparation of the manuscript.

**Competing interests:** The authors have declared that no competing interests exist.

areas dominated by people. These behavioural changes may function to reduce direct contact with diurnal human activity [6]. For example, tigers (*Panthera tigris*) and grizzly bears (*Ursus arctos*) have been shown to shift to nocturnal behaviour near people to minimize the risk of conflict and mortality [6, 7]. Such behavioural changes that decrease negative interactions may lead to coexistence, where humans and wildlife share landscapes with sustainable risks of death, injury, or significant cost to either party [8].

For human communities adjacent to natural (hereafter "wild") habitats, it is important to know whether local wildlife are adjusting to a range of human influences in a manner that can promote coexistence. Behavioural changes to avoid direct conflict may actually increase indirect conflicts due to attempts by animals to access calorie-rich human food attractants (e.g., accessing garbage, fruit, or crops at night when interaction with people is less likely). For example, increased nocturnality of cougars (*Puma concolor*) in human-dominated landscapes has been shown to increase their movement and caloric expenditure, and ultimately their potential to pursue anthropogenic attractants such as livestock [9]. Consequently, behavioural changes such as increased nocturnality could lead to either coexistence or conflict depending on context and conditions at the urban-wildlife interface.

Anthropogenic food attractants are a common source of human-wildlife conflict [1]. Attractants like garbage can appear beneficial as they are predictable, consistent, and spatially aggregated sources of possibly high-caloric food, thus requiring less foraging effort [10]. For potentially dangerous animals like large carnivores, their use of such attractants can provoke management responses that typically involve relocating or destroying the animal [11, 12]. Higher mortality in areas of greater conflict compared to wild areas with low or no conflict creates an ecological trap, whereby an animal selects the area based on its resources but experiences unexpectedly high mortality and lowered fitness [13]. These ecological traps function as population sinks which have negative impacts on long-term population persistence [14], and may become more common as the urban-wildland interface increases.

Coexistence with carnivores in the expanding urban-wildlife interface requires human mitigation to reduce wildlife visitation rates and mortality risks associated with human-dominated areas. Strategies include protecting anthropogenic food attractants or creating a hostile environment to signal the mortality risks to wildlife in human-dominated areas. For instance, if securing livestock indoors is not possible, setting up an electric fence to deter carnivores can reduce conflict [15]. Such actions may promote behavioural changes in carnivores to reduce conflict or maintain behaviours consistent with coexistence. Understanding the effectiveness of mitigation efforts and whether new strategies are needed requires knowledge about how carnivores use habitats and resources within the urban-wildland interface, and how that relates to the probability of conflict.

Black bears (*Ursus americanus*) often occupy urban-wildland interfaces in North America and have been found to change their natural behaviours as a result of human disturbance (including human presence and development) [10, 16]. They are a forest-adapted species with generalist diets and can alter their foraging patterns to access new food sources [10, 12, 17]. In the urban-wildland interface, omnivorous bears may select for anthropogenic over wild food sources [12], preying on livestock and consuming garbage, compost, bird feed, and fruit. The high caloric value and availability of anthropogenic food is especially attractive in the summer and autumn when bears need to store energy for hibernation, and can result in seasonal increases in human-bear conflicts [11, 18]. If such attractants remain constant and available, they can draw animals from surrounding wild areas and create a reliance on anthropogenic food, which may change bear behaviour in the urban-wildland interface. In Nevada, urban black bears are less active each day (as they satisfied their caloric needs faster), more nocturnal,

and hibernate less [10]. Evidence suggests that some bears do not hibernate at all in moderate climates with continuous food sources like garbage in the winter [19].

Previous research on black bear habitat use has found that bears typically avoid people, selecting forested habitat close to edges and at low elevations, where they can access vegetation in meadows and riparian areas [20, 21]. But by preferring edges, black bears are primed to thrive in the urban-wildland interface, as rural and suburban development creates more edge habitat which may increase natural foods like berries [22] and bring in new anthropogenic food attractants. On southern Vancouver Island in British Columbia, Klees van Bommel et al. [18] found that human-black bear conflict probability increased where suburban development adjoined wild areas, and in places with intermediate human density; but in autumn, high-conflict probability expanded into rural/agricultural areas, when reports of human-bear conflict shifted from being driven primarily by garbage attractants to include fruit, compost, and other rural attractants. It is thus likely that seasonal conditions affect the attractiveness of different types of human areas for bears, as is seen with the increase of rural conflicts during summer and autumn when crops ripen. Therefore, it is important to study bear habitat use both spatially and temporally to understand the implications for conflict.

## Study objectives

We aimed to understand the spatial and temporal dynamics of black bear habitat use along a gradient of human disturbance, from urban to wild, in the Capital Regional District of Vancouver Island, British Columbia (BC), Canada—a region reporting hundreds of human-black bear conflicts per year [18]. We conducted a camera trap study, and estimated habitat use by bears in urban, rural, and wild areas, as defined by human density and land use type. We measured and modelled the spatial distribution of black bear camera trap detections as a function of environmental and anthropogenic variables over a full year. To assess the relationship between habitat use and conflict, we subsequently modeled detections using previously estimated seasonal conflict probabilities from the same region [18]. Finally, we tested if bears were more active at night in human-dominated urban and rural landscapes compared to wild areas.

We tested two contrasting hypotheses to explain patterns of bear habitat use relative to reported human-bear conflict. Our "conflict hypothesis" posited that bear habitat use was driven primarily by available anthropogenic foods, such that bears would be most active in rural areas (i.e., the interface between urban and wild) where they had access to a broader spectrum of human attractants, such as garbage and bird feeders, but also unique attractants like livestock and food crops in addition to wild resources. Under this hypothesis, bears would not be expected to modify their behaviour to avoid conflict, such that habitat use would have a positive relationship with conflict probability and the degree of nocturnality would be similar in urban versus wild areas. An alternative "coexistence hypothesis" posited that bears changed their behaviour in the interface to avoid humans and thus avoid areas of high conflict probability, due to perceived costs (e.g. increased mortality risk) outweighing the potential benefits of anthropogenic resources [5, 10]. Under this hypothesis, we predicted that the areas of greatest habitat use by bears would be those with a low conflict probability, and that bears would be more nocturnal in urban and rural areas compared to wild areas.

## Methods

### Study area

Vancouver Island, BC, Canada, is home to black bears living at high densities near urban areas. Recent bear population estimates for Vancouver Island are not available, however high bear abundance is indicated by some of the highest average annual harvest densities across BC

during the past ten years (up to 25 bears/100 km$^2$) [23]. The municipality of Sooke, on the southern tip of Vancouver Island, has a current human census of approximately 13,000 that forms part of Victoria's Capital Regional District (CRD) population of 383,360 [24]. Sooke is the second fastest growing of 13 municipalities in the CRD, with growth rapidly converting forest into housing developments. The forested landscape around Sooke is comprised of second growth dry coastal western hemlock forests with Douglas-fir as the leading tree species, berry-producing shrub understories, and productive salmon rivers [25]. Between 2011 and 2017, Sooke had 60% of the reported human-bear conflicts in the CRD, and more than twice the number of calls to conservation officers than other municipalities. In that same timeframe, Sooke also had the highest conflict-related mortality for black bears, accounting for 38 of the 60 bear deaths reported in the CRD [British Columbia Conservation Officer Service, *unpublished data*].

## Camera traps

We set 54 camera traps within an 80 km$^2$ area in and adjacent to Sooke to assess spatial and temporal variation in bear distribution and habitat use along a gradient of human disturbance from urban to wild (Fig 1). Camera traps are an established non-invasive tool for monitoring large terrestrial animals with minimal human influence on behaviour, and are effective with clear research objectives [26, 27]. Based on estimated female black bear home ranges on Vancouver Island (7.83 km$^2$) [22], we expected the study area to include multiple overlapping home ranges, capturing a representative sample of the local bear population.

We deployed cameras following a stratified random design [28] to representatively allocate cameras based on the proportion of the survey area falling within each of three strata: urban (n = 11 cameras), rural (n = 19), or wild (n = 24). We aimed for >200 m between neighboring camera sites (mean = 446 m, range = 147–1467 m) to increase spatial independence [29]. Within strata, sampling distribution was randomized where possible. Due to the abundance of private land, urban and rural camera sites were selected from a candidate list of participating landowners provided by the local environmental non-governmental organization, Wild Wise Sooke. Rural sites were either within agricultural land cover or low development areas, while urban sites were in town and close to other homes. Wild sites were in forested areas with minimal disturbance from human development, consisting of 21 in Sea to Sea Regional Park and three on undeveloped T'Sou-ke Nation lands. Work was completed under a CRD Regional Parks park use permit (#15/19) and signed consent to entry forms from all private landowners and T'Sou-ke Nation. To randomize sampling locations within the main accessible block of the regional park, a 500 by 500 m grid was overlaid on park trail maps and cameras were placed in 10 random cells that contained a trail. The T'Sou-ke Nation forest sites and regional park sites on the northwest edge were only accessible by a single hiking trail, so cameras were set a minimum of 200 m apart. To avoid excessive human photos and privacy concerns, we avoided setting cameras directly on the main hiking trails in the park and T'Sou-ke Nation lands, and either targeted adjacent game and low-use human trails within the selected cell or set cameras off the main trail. Deployment occurred between July 18—August 20, 2018. To detect any seasonal variation in black bear habitat use [22], all cameras remained deployed for approximately one year, and were retrieved between July 16–19, 2019. We used a combination of three camera trap models (Reconyx PC900, Reconyx HC600, and Browning Strike Force HD Pro) randomly allocated across strata to reduce potential effects of different detectability between camera models.

We set cameras at locations to maximize the probability of detecting bears that occurred there, using local knowledge of where bears moved across urban or rural properties, or

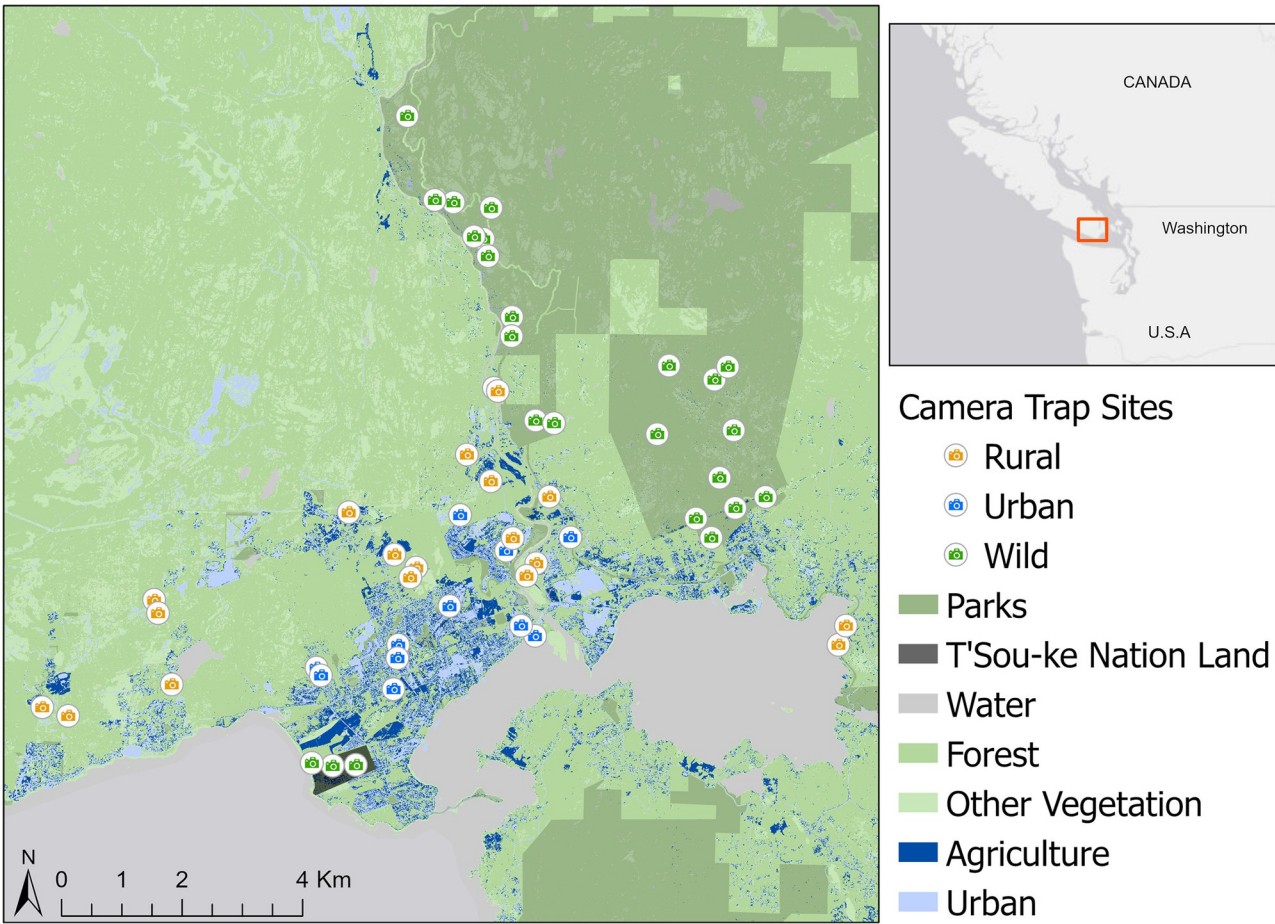

**Fig 1. Map of Sooke study area and camera trap locations.** Location of 54 camera traps used to sample black bear habitat use as measured by camera trap detection rates in and around the municipality of Sooke on Vancouver Island, BC, Canada. Other vegetation includes shrubs, herbs, wetlands, and shorelines outside of urban or agricultural land uses.

the presence of animal trails and sign. Per site, one camera was set on a tree, approximately one metre above the ground, at high sensitivity, with a one second delay between triggers (one image per trigger as bears are large enough to be captured without a sequence and this saves battery and memory card space) [30], and facing open spaces such as meadows, lawns, or trails. Black bears have shown a preference for using low-use human paths because of the ease of movement and increased shrub vegetation containing berries [22, 31]. Where possible, cameras faced an intersection of multiple animal and/or low-use human trails.

We visited camera traps every 2–3 months to download images, check functionality and replace batteries as needed. We used Timelapse Image Analyzer 2.0 [32] to classify all camera trap images of black bears. We defined independent detection events as those separated by ≥30 minutes to minimize correlation among consecutive detections as individual bears were not uniquely identifiable [27]. We counted sows with cubs as single individuals because sows determined the habitat use. We summed the number of detection events at each camera site for each month to calculate the monthly camera trap detection rate as a measure of habitat use (13 months x 54 cameras).

## Habitat use modelling framework

We used zero-inflated generalized linear mixed modelling (GLMM) to model bear habitat use. Our dependent variable was habitat use as measured by the monthly, camera-level detection rate of bears (i.e, count of independent detections). We chose to directly model detection rates instead of commonly proposed occupancy models as estimates from the latter may not be reliable for species with relatively large home ranges compared to the spacing of sampling points [33]. We used a zero-inflated approach in order to accommodate the high proportion of zeros (80%) due to some cameras not detecting bears in each month or at all [34]. To control for non-independence from repeat bear detections at a site across months, we included a random intercept for camera site in all models. All models also included the number of active camera days as an offset to account for variable sampling effort (i.e. not all cameras were active for all study days). Models were run with the '*glmmTMB*' package [35] in Program R [36]. We used the negative binomial distribution "Nbinom2" which treats the variance quadratically because all candidate models had a lower AICc compared to the "Nbinom1" distribution (where variance was treated linearly) [35].

**Habitat use predictor variables.** To relate bear detections to environmental and anthropogenic features associated with conflict probability, we considered a suite of camera-specific independent variables extracted from spatial datasets. We included human density, trail density, road density, elevation, and distances to agriculture and urban land cover (S1 and S2 Tables), averaged within a 150 m radius weighted buffer centred on camera locations in order to avoid overlapping buffers. These are the same predictors and buffers used in previous research to model human-black bear conflict in the CRD [18] to allow for direct comparison of their importance in explaining reported conflicts at the regional scale (CRD) versus bear habitat use at the local scale (Sooke). For variables derived from GIS raster layers with cells that extended beyond the buffer boundary, values were proportionally weighted to the cell areas within the buffer. Additionally, we used the Enhanced Vegetation Index (EVI) as a measure of vegetation productivity to indicate forage availability, rather than distance-to-forest because all camera locations were set within treed areas. We extracted EVI from MODIS 250m 16-day layers [37]. Unlike other common vegetation indices, EVI does not saturate in high biomass areas like forests, remaining sensitive to variation [38] and therefore informative across the forested sites. EVI has been used as a proxy for fruit abundance (grapes, *Vitis spp.*) in rural areas as ripeness peaks at the same time as greenness [39]. EVI may also serve as a proxy for season as greenness is highest in productive spring and summer months. We used a weighted average based on number of days the 16-day MODIS window had within our focal calendar month of analysis and the amount each raster cell fell into a 150m buffer around each camera site.

Additional predictors captured local-scale variation in natural food occurrence and recent conflict reports. We used distance-to-freshwater as a proxy for the documented importance of riparian vegetation and fish for black bears [40]; presence/absence of salmon at camera sites near (within 150 m buffer radius) salmon-bearing water by month [Charters Creek Hatchery, *unpublished data*], given their importance as a seasonal food resource for bears [22]; and the number of reported conflicts within a 500 m buffer of a camera site within the study year [41]. The buffer size for the latter two additional variables were tested at 150 and 500 m as in Klees van Bommel et al. [18].

We prepared continuous predictor variables by standardizing them, and ensuring noncollinearity ($r < 0.7$) [42]. Standardization by subtracting the mean and dividing by 1 standard deviation allowed for direct comparison of estimated effect sizes on bear habitat use. We computed the correlation matrix using Pearson correlation. We also tested for non-independence

in terms of spatial autocorrelation in the residuals from best-supported models (see below) using Moran's I test [43].

**Habitat use candidate models.** We specified a set of candidate models as competing hypotheses to explain spatial and temporal variation in black bear habitat use (Table 1). We expected if our overall "conflict hypothesis" was supported, the best fit model would show selection for human-dominated areas with high values for variables associated with high conflict probability; selection for wild areas with high values for low conflict probability variables, would support our "coexistence hypothesis". We tested four sets of candidate models to see which types of predictor variables best explained bear habitat use. The first model ("conflict") tested if bears were selecting areas with conditions that increased conflict probability, using the same predictors as Klees van Bommel et al. [18] used to model regional-scale on human-black bear conflict in the same study area, but applied to the local camera scale, with EVI to represent forest cover and vegetation productivity as a proxy for food and cover. The second model ("anthropogenic") tested if bears selected areas based primarily on human attractants and disturbance, by considering only anthropogenic variables and the recent conflict reports. A third model ("environmental") tested if bears were using areas based on natural food occurrence and security cover, by considering only environmental variables, including the distance-to-freshwater and presence of salmon variables. A fourth model ("full") was a full model with all conflict, anthropogenic and environmental predictor variables. The final model was a null model. We also included a quadratic term for human density in all candidate models with anthropogenic variables, to allow for bear selection of areas of intermediate density (i.e. rural areas), as well as an interaction between human density and trail density to test if bears used (or avoided) trails to navigate through human-dominated areas.

Candidate models for black bear habitat use as measured by monthly camera trap detection rates, from 54 camera traps sampled in and around Sooke, BC, Canada from July 2018 –July 2019 using zero-inflated GLMMs. Models are grouped by candidate set, with sub-models representing variations in variables (quadratic terms or interactions) included if they are within 2 ΔAICc of the best model in the set. Evaluated predictor variables extracted from a 150 m buffer around camera locations include HD = human density, RD = road density, EVI = enhanced vegetation index, DUrb = distance-to-urban, DAg = distance-to-agriculture, Con = conflict (500 m buffer), DW = distance-to-freshwater, Sal = salmon, Ele = elevation, and TD = trail density. All models also have site as a random effect and number of active days as an offset. Df is the degrees of freedom of the model, within ΔAICc is the difference in AICc scores from the top model within a set, between ΔAICc is the difference in top models between sets, Akaike weight is the relative likelihood of a model divided by the sum of those values across all models.

**Table 1. Black bear habitat use candidate models.**

| Hypothesis | Predictor Variables | df | Within ΔAICc | Between ΔAICc | AkaikeWeight |
|---|---|---|---|---|---|
| **Conflict** | HD + HD$^2$ + RD + EVI + DUrb + DAg + Ele + TD + TD*HD | 13 | 0.0 | 0.0 | 0.41 |
| | HD + HD$^2$ + RD + EVI + DUrb + DAg + Ele + TD | 12 | 0.6 | | 0.31 |
| **Full Model** | HD + HD$^2$ + RD + EVI + DUrb + DAg + Con + DW + Sal + Ele + TD + TD*HD | 16 | 0.0 | 2.7 | 0.44 |
| | HD + HD$^2$ + RD + EVI + DUrb + DAg + Con + DW + Sal + Ele + TD | 15 | 0.6 | | 0.32 |
| **Anthropogenic** | HD + HD$^2$ + RD + DUrb + DAg + Con + TD + TD*HD | 12 | 0.0 | 4.7 | 0.48 |
| | HD + HD$^2$ + RD + DUrb + DAg + Con + TD | 11 | 0.7 | | 0.34 |
| **Environmental** | EVI + DW + Sal + Ele | 8 | | 11.5 | |
| **Null** | | | | 24.1 | |

We used Akaike's Information Criterion corrected for small sample sizes (AICc) to assess statistical support among candidate models and selected the model with the lowest AICc. We also calculated the Akaike weight, which is the relative likelihood of a model divided by the sum of those values across all models [44], using "ICtab" from R package *bbmle* [45].

## Habitat use vs. seasonal conflict probability

To test if black bear habitat use had a positive relationship with conflict probability, and thus test predictions from a regional model of conflict at a local scale, we also modeled camera trap detections using the predicted spatially explicit probabilities of human-black bear conflict estimated from Klees van Bommel et al. [18]. Given that black bear habitat use is expected to vary seasonally following food availability, we used seasonal conflict probability and modelled an interaction with season (spring: February-April, summer: May-July, autumn: August-October, and winter: November-January). The months covered by each season have been adjusted to allow autumn to include the period of greatest berry abundance [22] and the timing of hyperphagia. Other predictor variables included season and strata (urban, wild, and rural).

We considered a set of 8 candidate models, which included each predictor modelled individually, 4 models in which season and strata were either additive or interaction terms with seasonal conflict probability, and a final null model (Table 2). Seasons were modelled as a factor, with spring used as the intercept. All models included site as a random effect and the number of active camera days as an offset to account for variable sampling effort. Each model was run using zero-inflation and used either the "Nbinom1" and "Nbinom2" variance structure–whichever resulted in a lower AICc. We used AICc to compare all models due to small sample sizes.

Candidate models for the relationship of seasonal black bear conflict probabilities with black bear habitat use as measured by monthly camera trap detection rates, from 54 camera traps sampled in and around Sooke, BC, Canada from July 2018 –July 2019 using zero-inflated GLMMs. Evaluated predictor variables include seasonal conflict probability extracted from 150 m buffer around camera locations from Klees van Bommel et al. [18]; seasons defined as spring: February-April, summer: May-July, autumn: August-October, and winter: November-January; and strata depending on where camera traps were set: urban, rural, or wild sites. All models also have site as a random effect and number of active days as an offset. Each model was run twice, using negative binomial 1 or 2 (nb1, nb2), and the model with the lower AICc is included below. Df is the degrees of freedom of the model, ΔAICc is the difference in AICc scores from the top model.

## Nocturnality

To test if Sooke black bears exhibited increased nocturnality in human areas as a potential indicator of coexistence behaviour, we compared the proportion of nocturnal bear activity

**Table 2. Black bear seasonal conflict probability candidate models.**

| Hypothesis | Predictor Variables | Distribution | df | ΔAICc |
|---|---|---|---|---|
| **Conflict * Season** | Seasonal Conflict Probability + Season + Conflict Probability Seasonal*Season | nb1 | 11 | 0.0 |
| **Conflict + Season** | Seasonal Conflict Probability + Season | nb1 | 87 | 3.5 |
| **Season** | Season | nb1 | 7 | 4.5 |
| **Conflict + Strata** | Seasonal Conflict Probability + Strata | nb1 | 9 | 29.7 |
| **Conflict * Strata** | Seasonal Conflict Probability + Strata + Seasonal Conflict Probability*Strata | nb1 | 5 | 31.5 |
| **Conflict** | Seasonal Conflict Probability | nb1 | 6 | 34.4 |
| **Strata** | Strata | nb2 | 4 | 79.5 |
| **Null** | | nb2 | | 83.6 |

between areas of high (urban and rural sites) vs. low disturbance (wild sites) [5]. We first classified all independent bear detections as either diurnal (between sunrise and sunset) or nocturnal (between sunset and sunrise), using the R package *suncalc* [46]. We then calculated risk ratios (RR) and associated variance for the urban and rural categories as

$$RR = \ln\left( X_h / X_l \right)$$

$$Variance(RR) = \left( \frac{1}{O_{High,night}} - \frac{1}{O_{High}} \right) + \left( \frac{1}{O_{Low,night}} - \frac{1}{O_{Low}} \right)$$

where $X_h$ is percent nocturnal activity (i.e., the proportion of night detections out of all detections) at high disturbance (e.g, urban or rural), $X_l$ is percent nocturnal activity at low disturbance (e.g., wild), and O is the number of observations [5]. RR > 0 would suggest a greater degree of nocturnality in response to a greater human presence, with larger numbers representing greater nocturnality, whereas a RR < 0 would indicate lower nocturnality in response to increased human presence. We assessed support for changes in nocturnality by comparing the observed RR values to a bootstrapped distribution of RR values, which was created by randomly assigning detections into urban/rural/wild categories 1000 times and calculating RR values. We considered changes in nocturnality to be significant if the 95% confidence intervals (CIs) of observed RRs did not overlap with the 95% highest posterior density intervals (HPDIs) of the bootstrapped RR distributions. We used the "HDInterval" [47] package in R.

## Results

Camera traps detected black bears in all 13 months of the survey (N = 548 independent detections over 16,546 camera trap days), averaging 42 monthly detections across all sites, with the most detections in September 2018 (n = 148) and the least in February and March of 2019 (n = 1 each; Fig 2 providing evidence that not all bears hibernate). Rural sites had the most detections (n = 368), followed by wild and urban sites with 103 and 77 detections respectively (S4 Table) [48]. While individual bears are not reliably distinguishable from camera trap images, we did photograph multiple bears of different sizes at the same sites, sows with cubs of different sizes taken around the same date, and multiple photos taken close in time at different sites that were unlikely to be the same bear.

Patterns of habitat use by black bears in Sooke were best explained by the "conflict model", which included anthropogenic and environmental variables that explained broader patterns of conflict in the CRD (Akaike weight: 0.41; Table 1, full model set S3 Table). However, the direction of effects did not fully match the predictions of either our "conflict" or "coexistence" hypotheses. Bears were more active in areas with greater EVI and higher vegetation productivity (0.34 ± 0.13, p < 0.01; 95% confidence intervals do not overlap zero, Fig 3), which was low-conflict parks and crown land forest in winter and spring, but high-conflict rural areas in summer and autumn (S1 Fig). We failed to detect an effect of human density (effect size: -2.37 ± 1.85 standard error, p = 0.20; quadratic term: -0.49 ± 0.25, p = 0.05), road density (-0.22 ± 0.24, p = 0.34), elevation (-0.52 ± 0.27, p = 0.06), distance-to-urban (0.17 ± 0.21, p = 0.41) and -agriculture (-0.12 ± 0.30, p = 0.68), or trail density (-2.31 ± 1.24, p = 0.06). While the "conflict model" was > 2 ΔAICc from the next hypothesis model set (the full model), a version within the "conflict model" set that did not include the interaction between trail and human density was only 0.6 ΔAICc away, suggesting this interaction did not add much to the explanatory power of the top model (-3.54 ± 2.25, p = 0.12). There was no evidence of spatial autocorrelation in the model residuals (Moran I = -0.052, p-value = 0.999).

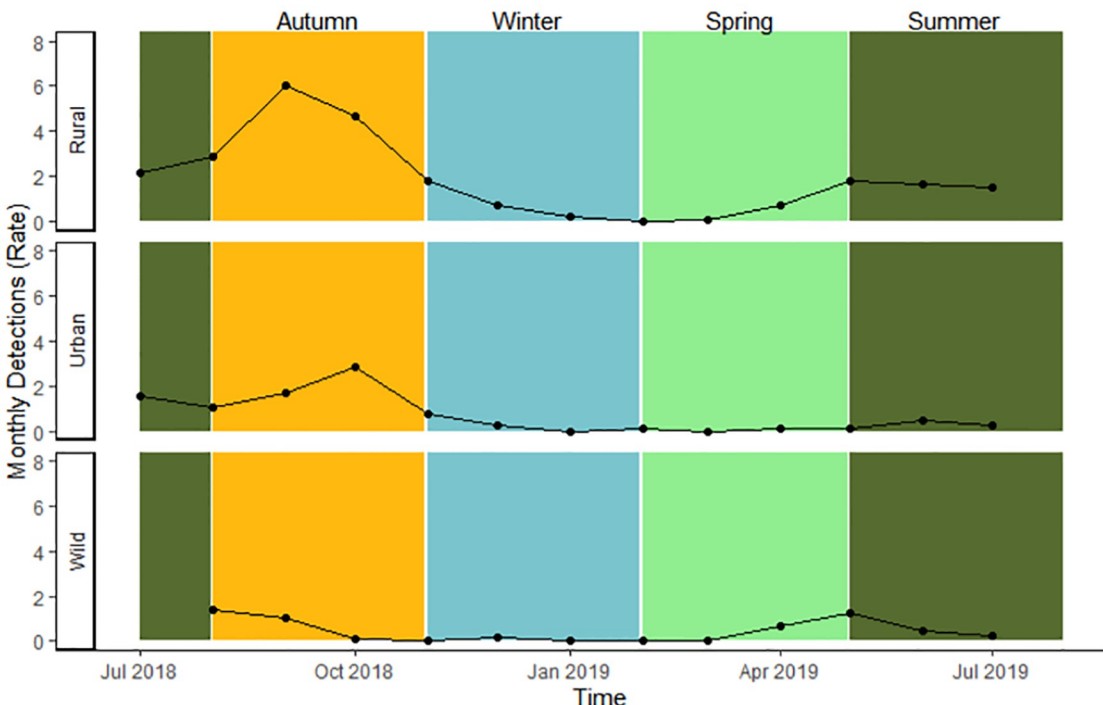

**Fig 2. Graph of monthly black bear detection rate.** Monthly black bear detection rate from 54 camera traps sampled July 2018–2019 in Sooke, BC, Canada grouped seasonally. The detection rate accounts for time when camera traps were not active, with rate = (number of detections / number of active camera days) * number of days in the month. Seasons are spring: February-April, summer: May-July, autumn: August-October, and winter: November-January.

Bear habitat use in the seasonal conflict probability model set was best explained by the model with an interaction between seasonal conflict probability and season (Table 2). Monthly bear detection rates (Fig 2) were significantly higher in summer (effect size: $1.37 \pm 0.52$ standard error, S.E., $p < 0.01$) and autumn ($1.12 \pm 0.47$, $p = 0.02$) compared to spring, while winter detections were similar to spring (-$0.27 \pm 0.52$, $p = 0.60$; Fig 4). Bear detections were generally lower in areas with increasing conflict probability (-$0.22 \pm 0.11$, $p = 0.05$), but less so in summer ($0.18 \pm 0.10$, $p = 0.09$) and winter ($0.22 \pm 0.11$, $p = 0.04$) compared to spring due to the interaction between conflict probability and season. By contrast, detections of bears in autumn were higher in areas with higher conflict probability ($0.22 \pm 0.11$, $p = 0.04$; Figs 4 and 5).

Consistent with the prediction of our "coexistence hypothesis", the risk ratios for nocturnal activity in urban versus wild areas ($RR_{urban} = 0.84$; 95% CI: 0.42–1.26) and rural versus wild areas ($RR_{rural} = 0.74$; 95% CI: 0.37–1.12) were both positive and did not overlap the bootstrapped 95% HPDIs (urban: -0.36–0.32; rural: -0.21–0.27); Fig 6, S4 Table), suggesting greater nocturnality in areas of higher human disturbance. Small sample sizes prevented estimation of season-specific risk ratios.

## Discussion

Our results suggest that black bears in Sooke change their behaviours in response to human presence. The municipality of Sooke has the highest reported rates of human-bear conflict and conflict-related bear mortality on Vancouver Island, which is already known for having one of the highest rates of human-bear interactions in North America [49]. Therefore, the Sooke area represents an opportunity to investigate whether bears are changing their behaviours in

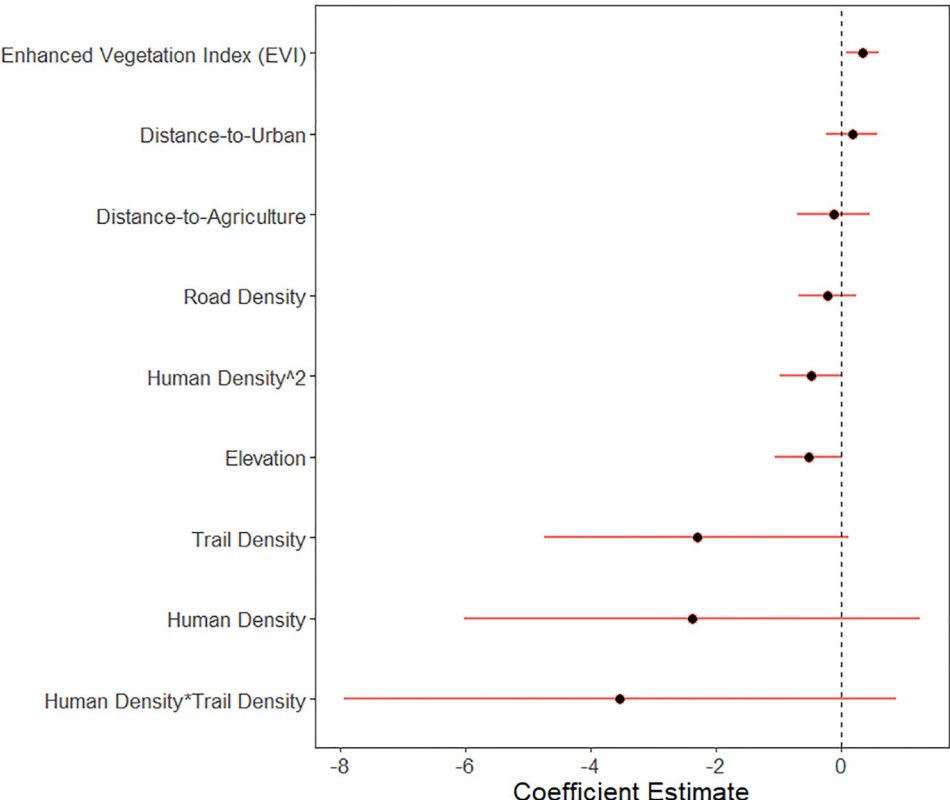

**Fig 3. Black bear habitat use model results.** Estimated effects of human and environmental variables on black bear habitat use as measured by camera trap detection rate in Sooke, BC, Canada. Variables are human, road, and trail density (including a quadratic variable for human density and interaction between human and trail density), distance-to-agriculture and -urban, enhanced vegetation index, elevation, and camera trap active days. Coefficients from best-supported zero-inflated negative binomial generalized linear mixed model of monthly detections from 54 camera traps sampled July 2018–2019 illustrated as mean and 95% confidence intervals. Predictor variables have been standardized to a mean of zero and standard deviation of one to allow for direct comparison.

response to humans in ways that either facilitate coexistence or result in conflict with people. Our results showed that the conflict model best fit the camera detection data, with vegetative condition significantly influencing bear habitat use. Furthermore, the relationship between camera trap detections and previously modelled probabilities of conflict shows that bears are shifting their habitat use spatially by season and increasing their nocturnal activity in human spaces.

Our results suggest, however, that bear responses to conditions in the urban-wildland interface are not consistent year-round and do not support simple interpretations of conflict versus coexistence. Increased nocturnality in urban and rural spaces compared to wild areas and preference for vegetatively productive habitats (high EVI) suggest coexistence for much of the year, but increased use of high-conflict rural areas still occurs in autumn. This is similar to observations of seasonal shifts in natural diets before hyperphagia, where bears select swamp habitat with an abundance of berries, grasses, and willows [50]. We suggest that autumn conflict reflects the bears' hyperphagia to prepare for winter denning, coupled with the peak availability (ripeness) of anthropogenic crops, led to riskier bear behaviour.

Spatially, bears used habitats with higher EVI, suggesting they were selecting coniferous forest for shelter and natural foods in winter and spring [20, 21]. This could be a result of bears

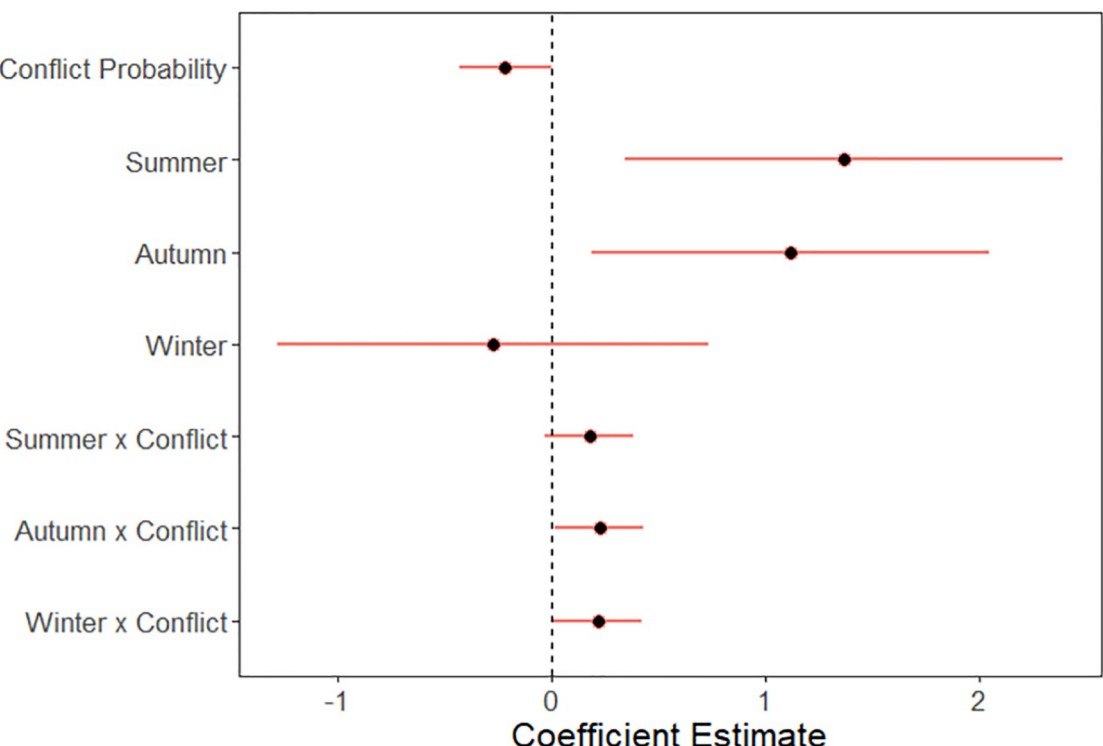

**Fig 4. Black bear seasonal conflict probability model results.** Estimated effects of seasonal conflict probability and season on black bear habitat use as measured by camera trap detection rate in Sooke, BC, Canada. Coefficients from best-supported zero-inflated negative binomial generalized linear mixed model of monthly detections from 54 camera traps sampled July 2018–2019 illustrated as mean and 95% confidence intervals. Seasons are spring: February-April, summer: May-July, autumn: August-October, and winter: November-January. Predictor variables have been standardized to a mean of zero and standard deviation of one to allow for direct comparison.

avoiding people as predicted by our "coexistence hypothesis", which may thereby minimize conflicts. While, our model results did not find a significant effect of human density on habitat use, previous studies in the North Cascades in Washington (USA), and Michigan (USA), found that black bears avoided people based on metrics of developed lands and roads [20, 51]. Interestingly, road density and distances to urban and agricultural areas were also not significant predictors in our study, while edge habitats (i.e., those close to disturbed areas) have been found to be selected by bears in the Cascade Mountains, USA [21]. The lack of a similar effect in our study may have resulted from the 150 m buffer size we extracted our variables at, or suggest that bears in Sooke select habitat based on food availability and avoidance of direct human encounters rather than general landscape condition. Future research should further evaluate the importance of avoiding direct human encounters relative to responding to habitat changes such as landscape disturbance and supplemental human food resources for bears in Sooke.

Bear activity was significantly more nocturnal in urban and rural areas than at wild sites, further suggesting they were avoiding direct interactions with humans and possibly facilitating human-wildlife coexistence [5]. However, nocturnal activity could increase a bear's ability to access food attractants like garbage, compost, or livestock, which may be less guarded by people at night. This is supported by research from Colorado which found that bears became more nocturnal in years of poor natural food availability, as they used areas of higher human density to access anthropogenic food sources [52]. Furthermore, nighttime activity in bears

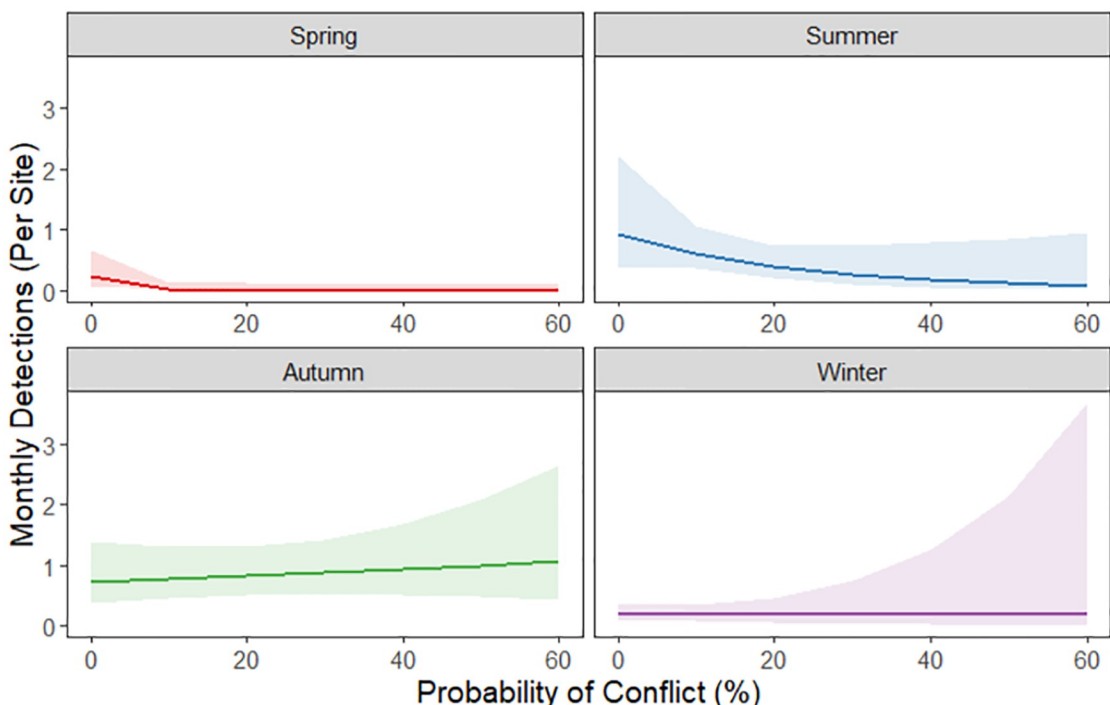

**Fig 5. Graph of black bear detections by season vs. probability of conflict.** Plot of relationship between monthly black bear camera trap detections and the interaction between seasonal conflict probability and season (spring: February-April, summer: May-July, autumn: August-October, and winter: November-January) represented by mean and 95% confidence intervals.

has been found to peak in spring, when food is scarce, and in autumn, when energetic requirements are high, further suggesting that increased nocturnality is an adjustment to facilitate access to human resources [53]. Therefore, despite the change to nocturnal behaviour, conflicts could still be detrimental to bear fitness if they cause property damage property that results in reactive management of conflict bears.

Overall, bear avoidance of humans in the Sooke study area fosters coexistence in the urban-wildland interface for most of the year, with notable exceptions in the late summer and fall when a combination of hyperphagia and increased availability of high-quality anthropogenic food resources likely increase bear tolerance of human presence and attract bears into rural conflict. Consistent with our "conflict hypothesis" bears avoided high conflict areas less in summer (May-July) and even used them more in autumn (August-October). At these times of year, high conflict areas are largely human-dominated rural areas, which have the broadest spectrum of anthropogenic food attractants available. If our assumption that higher EVI indicates forage availability and crop ripeness is correct, as has been shown in other studies [12, 38, 39], Sooke bears may be selecting for crops, orchards, and berries along forest edges that provide important, abundant, and aggregated foods during hyperphagia. Fruit especially has been found to be selected over natural and other anthropogenic attractants by black bears in Nevada [12] and grizzly bears in Canada [13]. Bears may then be drawn into human areas as they are attracted to fruits and crops coming into season. Indeed, the conflict probability model showed that bear detections increased in autumn despite increasing conflict probability. Therefore, bear habitat use appears to reflect a spatiotemporal shift made to balance increased caloric needs with risk of conflict.

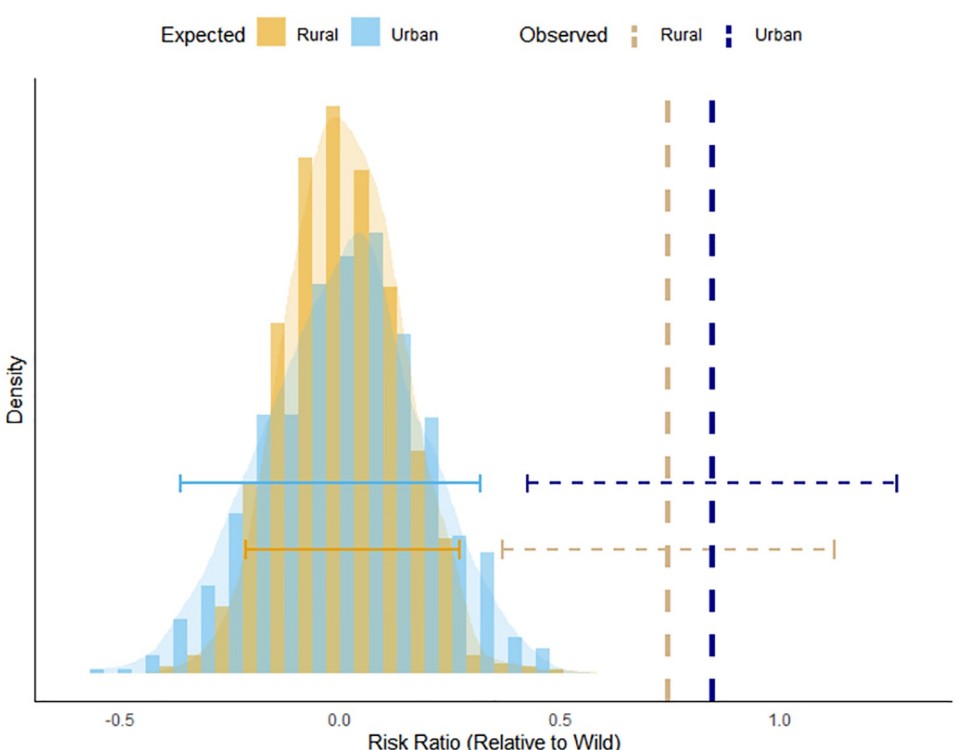

**Fig 6. Graph of observed black bear nocturnal activity in urban and rural areas vs. expected.** Expected null distribution versus observed risk ratios for black bears nocturnality from urban and rural camera trap sites relative to wild sites. The risk ratio (RR) compares nighttime activity for bears detected in areas of high human disturbance ($X_h$, urban = blue, rural = yellow) with those in low disturbance ($X_l$, wild) areas using the equation Risk Ratio = $\ln(X_h/X_l)$, where 0 would mean no difference. Expected distribution was calculated using 1000 bootstrap iterations from data. Horizontal lines show the 95% Highest Posterior Density Intervals for the expected (null) distribution (solid lines) and 95% confidence intervals for the observed values (dashed lines).

We do note that our second set of models comparing habitat use to seasonal conflict probability did not propagate the uncertainty from the models used to estimate the conflict probabilities. Future research could further integrate reports of human-black bear conflict with camera trap surveys to more accurately predict and target where and when conflicts may happen, as in Fidino et al. [54]. Conflict reports themselves are a sample of all the conflict that occurs, and thus contain error as some conflict goes unreported. However, community demographics have been found to have limited influence on the chance of reporting conflicts, and conflicts relating to safety or property damage (which encompasses many human-black bear conflicts) are more likely to be reported [18]. We therefore assumed that sampling of conflicts across Sooke was not systematically biased.

Current mitigation efforts may not be enough to promote coexistence by reducing attractants or increasing bears' perceived risk of human areas. While the spike of rural conflicts in autumn may result from fruit and other rural attractants, garbage conflicts are the focus of current mitigation priorities because they are the most common type of human-bear conflict year-round in Sooke [55]. We recommend that conflict mitigation strategies in Sooke should extend beyond urban and rural garbage management to include managing autumn attractants, such as by picking fruit in a timely manner and keeping attractants such as compost or crops within electric fences or other bear proof containers.

In addition to mitigation, land managers should also prioritize conserving or restoring natural food sources. Black bears have been found to use human food attractants more when there was a natural food shortage, and then revert back in subsequent years as natural foods recovered [52]. Protecting natural resources may gain importance as climate change affects the range, abundance, and timing of foods like berries [56] and salmon [57].

Our study provides a foundation for future research on human-bear interactions in this and similar hotspots of conflict. Useful next steps would be to estimate bear densities in the urban-wildland interface [58], and monitor trends over time in comparison to wild regions. Furthermore, to determine if Sooke is an ecological trap for bears, tracking individual movements could determine if bears are drawn to urban spaces from wilder areas, and whether conflict-related deaths are affecting the broader population, or if a few "problem" individuals are responsible for the majority of conflicts [13]. If bears are attracted to urban areas, determining the sexes of those individuals would be important as females of reproductive age have a greater impact on population demography. In general, male black bears have been found to use more developed areas and thus be involved in more conflict [12, 16], however in poor natural food years, females may select areas of higher development than males [59]. Extending the sampling period of the current design while adding finer scale data on anthropogenic and natural bear foods–particularly salmon and berries–could also provide information on how dependent these bear conflict behaviours are on natural food availability, and whether natural or anthropogenic foods are supporting bears who do not stay in dens over winter.

Coexistence, as defined by Carter and Linnell [8], occurs when humans and wildlife share landscapes without unsustainable risk of death, injury, or significant cost to either party. As the urban-wildland interface expands, coexistence will require both humans and wildlife to adapt to avoid conflict. The current combination of black bear use of low-conflict, forested habitats for most of the year, and their increased nocturnal behaviour in areas of higher human density, represents behavioural plasticity via spatial and temporal avoidance of humans, thereby reducing the chance of dangerous conflicts and contributing to human-bear coexistence [6]. However, bear use of urban areas and their attraction to rural areas in summer and autumn suggest that bears are not entirely avoiding risks associated with food-related conflicts. As long as seasonal rural conflicts continue, bears may be targeted for lethal mitigation or become food-conditioned and desensitized to humans, and thus be at risk of escalating dangerous conflicts. Thus, coexistence with bears and other large carnivores necessitates some human tolerance of conflict risk, with an aim of reducing that risk to an acceptable level through mitigation strategies and practices [8]. Mitigations will vary by species and socio-ecological context; our study shows how research can determine the patterns of space use by wildlife in human-dominated areas and relate them to conflict outcomes to generate context-specific management recommendations.

## Supporting information

**S1 Table. Black bear habitat use predictor variables.** Predictor variables used to model black bear habitat use in Sooke, Vancouver Island, Canada, between 2018–2019. Variables all derived within 150 m radius buffers around camera trap locations unless otherwise noted. Weighted buffers reduce the contribution of raster layer cells not fully within the circular buffer by the percent excluded.
(DOCX)

**S2 Table. Mean values of black bear habitat use predictor variables.** Mean values of predictor variables by strata used to model black bear habitat use in Sooke, Vancouver Island,

Canada, between 2018–2019.
(DOCX)

**S3 Table. Complete set of black bear habitat use candidate models.** All candidate for black bear habitat use as measured by monthly camera trap detection rates, from 54 camera traps sampled in and around Sooke, BC, Canada from July 2018 –July 2019 using zero-inflated GLMMs. Evaluated predictor variables extracted from a 150m buffer around camera locations include HD = human density, RD = road density, EVI = enhanced vegetation index, DUrb = distance-to-urban, DAg = distance-to-agriculture, Con = conflict (500 m buffer), DW = distance-to-freshwater, Sal = salmon, Ele = elevation, and TD = trail density. All models also have site as a random effect and number of active days as an offset. Df is the degrees of freedom of the model, within ΔAICc is the difference in AICc scores from the top model within a set, between ΔAICc is the difference in top models between sets, Akaike weight is the relative likelihood of a model divided by the sum of those values across all models.
(DOCX)

**S4 Table. Count of day vs. night black bear detections.** Number of independent black bear detections in the day versus night at urban, rural, and wild camera trap sites (n = 548).
(DOCX)

**S1 Fig. Graph of average monthly Enhanced Vegetation Index in urban, rural, and wild areas.** Enhanced Vegetation Index (EVI) averaged within sampling strata (urban, rural, or wild) across 54 camera-trap sites in Sooke, BC, Canada sampled from July 2018–2019.
(TIF)

## Acknowledgments

We thank Peter Arcese and Garth Mowat for feedback on earlier drafts, Wild Wise Sooke for assistance finding land owners to work with, T'Sou-ke Nation for insight and allowing us to set cameras on their land, the District of Sooke for access to GIS data, and Mike Badry and the British Columbia Conservation Officer Service for access to their conflict reporting data. We thank Todd Golumbia, Joanna Burgar, Jacqui Sunderland-Groves, Erin Tattersall, Aisha Uduman, Alexia Constantinou, Taylor Justason, Meghna Bandyopadhyay, and Paige Monteiro for field assistance, and Emily Siemens, Lauren Kasper, Zach Brunton, Avril Hann, Micaela Anguita, HyunGu Kang and Isla Francis for help processing photos. We also thank our reviewers Stijn Verschueren and Mason Fidino for their constructive suggestions.

## Author Contributions

**Conceptualization:** Joanna Klees van Bommel, A. Cole Burton.

**Data curation:** Joanna Klees van Bommel.

**Formal analysis:** Joanna Klees van Bommel, Catherine Sun.

**Funding acquisition:** Joanna Klees van Bommel, A. Cole Burton.

**Investigation:** Joanna Klees van Bommel.

**Methodology:** Joanna Klees van Bommel, A. Cole Burton.

**Project administration:** Joanna Klees van Bommel.

**Resources:** Melissa Todd.

**Supervision:** A. Cole Burton.

**Validation:** Joanna Klees van Bommel, Catherine Sun.

**Visualization:** Joanna Klees van Bommel.

**Writing – original draft:** Joanna Klees van Bommel.

**Writing – review & editing:** Joanna Klees van Bommel, Catherine Sun, Adam T. Ford, Melissa Todd, A. Cole Burton.

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
