## [Decision Letter · Decision Letter 0]

4 Jul 2022

PONE-D-22-12898Coexistence or conflict: black bear habitat use along an urban-wildland gradientPLOS ONE

Dear Dr. Klees van Bommel,

Thank you for submitting your manuscript to PLOS ONE. After careful consideration, we feel that it has merit but does not fully meet PLOS ONE’s publication criteria as it currently stands. Therefore, we invite you to submit a revised version of the manuscript that addresses the points raised during the review process.

We look forward to receiving your revised manuscript.

Kind regards,

Bogdan Cristescu

Academic Editor

PLOS ONE

Academic Editor's (Bogdan Cristescu) comments:

Two reviewers provided excellent comments and suggestions, with the feedback from one of them resembling more of a major than minor revision. Please address their points in the revised manuscript and response to reviewers document.

It will be important to justify why an occupancy modeling approach was not used. One reviewer provided suggestions on some ways to achieve that.

Please include metrics of model fit throughout, for example in Tables 1 and 2.

Looking forward to seeing the revision.

Journal Requirements:

Natural Earth (public domain): http://www.naturalearthdata.com

Reviewers' comments:

Reviewer's Responses to Questions

**Comments to the Author**

1. Is the manuscript technically sound, and do the data support the conclusions?

Reviewer #1: Yes

Reviewer #2: Yes

2. Has the statistical analysis been performed appropriately and rigorously? 

Reviewer #1: Yes

Reviewer #2: No

3. Have the authors made all data underlying the findings in their manuscript fully available?

Reviewer #1: No

Reviewer #2: Yes

4. Is the manuscript presented in an intelligible fashion and written in standard English?

Reviewer #1: Yes

Reviewer #2: Yes

5. Review Comments to the Author

Reviewer #1: The authors analyze camera detection rates of black bears in and around a conflict hotspot. Urban-wildlife interfaces are expanding rapidly and understanding habitat use of key species in human environments is critical for their management. The manuscript is well-written, the objectives are clearly defined and addressed, and the findings are discussed within a relevant context. I have few comments:

The authors acknowledge complexities with individual bear identification and discuss future directions of research that include density estimates and individual movements. Is there however any indication that multiple and enough individuals have been captured during your study to be representative for the broader population. In particular because generalist species typically display a great amount of behavioral plasticity, and the extent of the study area is rather small (~80km2).

Related to above comment and a potential avenue for future research, would there be any (expected) differences in habitat use between adult males, adult females, and females with cubs? And how would this influence conflict or coexistence?

How did you deal with the fact that not detecting a bear at a camera trap location may be a false absence? And consequently the zero’s in the zero inflated model may not all be true absences, which may induce bias in inferences on habitat use?

L388-393: It is not clear to me on what base you make these inferences. You report, for example, a negative association between bear habitat use and human density, but no effects of trail density are reported. From looking at Fig3, it seems however that both variables have similar coefficient estimates, standard errors (not overlapping zero) and confidence intervals (overlapping zero). The same applies for the quadratic term of human density and elevation. I would therefore think that only ‘Active Days’ and ‘EVI’ are informative predictors of bear habitat use.

L414-418: Please be consistent in reporting p-values and could you include these in the previous results section?

L416: maybe add ‘compared to spring’.

L460-463 & L469-471: Is there any previous research in natural systems that shows seasonal shifts in habitat use as a consequence of hyperphagia?

Reviewer #2: # Review for:

*Coexistence or conflict: black bear habitat use along an urban-wildland gradient*

Note: This was written in markdown format. I've also included a PDF if that is easier for you to read.

In this paper the authors look at spatio-temporal variation in the distribution of black bears in an area where conflict between black bears and humans is high. Additionally, they also look at the correlation between black bear detections and the probability of black-bear conflict that was estimated from previous research. In general, I think this is an interesting and well written piece of research. I often found myself in agreement with the logical flow of the research, the modeling choices made, and the interpretation of said models. Great work!

Perhaps my biggest concern is the secondary modeling approach, which treats the estimated conflict probability as known, when in fact it is something that is estimated with error. In my review below I have a number of hopefully helpful citations about this specific issue for the authors.

I also suggest the authors look at a recent article of mine (Fidino et al. 2022), which shows how integrated models could be use to combine human-wildlife conflict data with camera trap data. I'm not on the hunt for a citation here, but since it is relevant to this piece of research I am sharing in case you were interested in how to do this in the future.

```

Fidino, M., Lehrer, E. W., Kay, C. A., Yarmey, N. T., Murray, M. H., Fake, K., ... & Magle, S. B. (2022). Integrated species distribution models reveal spatiotemporal patterns of human–wildlife conflict. Ecological Applications, e2647.

```

If you have any specific questions about any of my comments, feel free to email me at mfidino@lpzoo.org. Again, great work on this!

- Mason Fidino

## Abstract

---

### Line by line comments

Line 46: The 'such as' breaks up this sentence in a weird way. I think you can remove it and the examples become a little more clear.

## Introduction

---

### Top-level thoughts

1. Well written introduction, I only have some minor comments on some of the wording / sentence structure.

### Line by line comments

Line 64: Is sustainable the correct word here? I think it could be, but also it tends to have a bit of a positive connotation to it, so it reads a little off to say sustainable risks of death. Maybe 'acceptable' could be used instead? Just a minor point, feel free to ignore if you disagree.

Line 69: This sentence is a little circular. Changes to avoid direct conflict can increase direct conflict when risk of direct conflict is lower. I think it's the second use of direct conflict, which makes it feel a little redundant. Could that bit right after the first (e.g.,) just get removed? I think the point still comes across.

Line 76-77: Is 'year round' needed when you already state they are available consistently? "...as they are predictable, consistent, and spatially aggregated sources of..."

Line 82-84: Does it help here to remind the reader that the urban-wildlife interface is increases, and therefore increased risk of ecological traps for such species?

Line 88: ...to signal wildlife mortality risks...

Line 98: ...and can alter their foraging patterns to...

Line 103: ..., and can result in seasonal...

Line 103-105: Sentence starting with "If such attractants" reads a little weird. Maybe it's the 'attractants can attract animals' part or that the '...and food-conditioning' bit at the end feels like a different thought tacked on.

Line 105 - 108: This sentence could be made more active (and therefore a little more succinct). "In Nevada, urban black bears are less active each day (as they satisfied their caloric needs faster), more nocturnal, and hibernate less."

Line 116-120: Conflict probability of what? Do you mean human-wildlife conflict probability in general? Human-bear conflict in particular?

Line 123-125: Great point, and the introduction sets a really strong foundation for the need to study this.

## Methods

---

### Top-level thoughts

1. I would imagine that the distance between camera trapping locations would be species specific to maintain spatial independence. For black bear, multiple camera traps can be contained within the home range of a single bear, and therefore I would assume that spatial independence is not met. This citation is also a pre-print, though it's been around for 12 years. Maybe just soften the language here (e.g,. say to increase spatial independence rather than maintain spatial independence).

2. Was the sampling effort term included as an offset to the model or did you just include it as a covariate? If the latter was done, I would encourage adding a log offset term instead, as that is a more standard approach when there is variation in sampling effort. If I had to guess, you used a log-offset, so just be specific about that here in the methods.

3. I could see some readers get their hackles up about not using an occupancy model here. I leave it to the authors, but it may help to just get ahead of it here (by using a zero-inflated model you may be estimating habitat use conditional on presence anyways, depending on the class of zero-inflated model you fit). Likewise, occupancy models are literally just zero-inflated logistic regression, so your approach has substantial overlap. For example, the old MacKenzie et al. occupancy modeling book talks about these similarities on about page 135 (i.e., the use of the zero-inflated binomial to model occupancy).

```

MacKenzie, D. I., Nichols, J. D., Royle, J. A., Pollock, K. H., Bailey, L. L., & Hines, J. E. (2017). Occupancy estimation and modeling: inferring patterns and dynamics of species occurrence. Elsevier.

```

4. There are a few different kinds of zero-inflated glmms, can you be a little more specific? For example, what is the error distribution that was used (negative binomial, Poisson, etc.). Is this a two part model and therefore the conditional model cannot include zeroes or is it a mixture model and therefore the conditional models can include zero? A little more explanation here would be helpful. note: I see now that the distributional information is shared much lower, around line 297. I'd move that little bit of info upwards so it's not separated from when you introduce the modeling framework.

5. Thank you for using a set of candidate models to assess your different hypotheses. Solid approach. Given the two set hypotheses brought up in the introduction, in may help to add a little more connection between those hypotheses and the set of candidate models given that there are two hypotheses but four candidate models. You should also fit a fifth null model as well (just the active days term plus the site random effect). Given the results I suspect the null will provide the worst fit (highest delta AIC), but it's nice to demonstrate this.

6. Based on the introduction (lines 133-134) I thought that modeled conflict probabilities would be incorporated into your glmms, but instead it looks like counts of conflict are included instead. I suspect other readers will also be confused about this. What makes this more confusing, is that there is a second batch of models done that uses the estimated conflict probabilities.

7. Why conflicts over the year instead of conflicts per month?

8. Using the output from one model as a predictor in another is okay, but the uncertainty of those estimates should also be propagated into the secondary model. From my reading of this secondary model set, I'm guessing that these spatially explicit probabilities are treated as known (i.e., measured without error). If this is the case, using such predictions in secondary analysis leads to anticonservative tests because this error is excluded from further tests (i.e., estimates are too precise). Some papers about this topic that the authors may find useful include:

```

Hadfield, J. D., Wilson, A. J., Garant, D., Sheldon, B. C., & Kruuk, L. E. (2010). The

misuse of BLUP in ecology and evolution. The American Naturalist, 175(1), 116-125.

Houslay, T. M., & Wilson, A. J. (2017). Avoiding the misuse of BLUP in behavioural

ecology. Behavioral Ecology, 28(4), 948-952.

Link, W. A. (1999). Modeling pattern in collections of parameters. The Journal of

wildlife management, 1017-1027.

```

The Houslay & Wilson paper is open access, so that is where I'd start. I've personally found this easiest to account for in a Bayesian framework (e.g., if you have the mean and SE of each prediction you can set a prior for each data point to propagate that uncertainty), but there are likely ways to deal with this in a frequentist framework as well (e.g., bootstrapping, but resampling the predicted covariate instead of the response variable).

9. What is a reported conflict in these data? Do they vary in severity?

## Results

---

### Top-level thoughts

1. Failing to detect an effect does not mean that there was no effect (e.g., line 391 - 392). I'd just reword to "We failed to detect an effect of road density" so that you avoid confirming the null (which these tests do not do). Other than that, great breakdown of the results.

## Discussion

---

### Top-level thoughts

1. Given the conflicting hypotheses, do the authors feel that one hypothesis was supported more than the other?

2. Any caveats worth bringing up here? For example, there was the assumption that EVI indicates forage availability. Is it possible for there to be human-bear conflicts that go unreported and so the conflicts / year metric used may have some error?

### Line by line comments

Line 530: You used ecological trap earlier.

## Tables & figures

---

### Top-level thoughts

1. The axis text on many of the figures is a very light gray, I'd suggest replacing with black to make it easier to read.

2. You could increase the line width for the 95% CI's on figure 3&4, plus the mean estimate on figure 5.

6. PLOS authors have the option to publish the peer review history of their article (what does this mean?). If published, this will include your full peer review and any attached files.

Reviewer #1: **Yes: **Stijn Verschueren

Reviewer #2: **Yes: **Mason Fidino

---

## [Author Response · Author response to Decision Letter 0]

14 Sep 2022

Academic Editor's comments:

Two reviewers provided excellent comments and suggestions, with the feedback from one of

them resembling more of a major than minor revision. Please address their points in the

revised manuscript and response to reviewers document.

It will be important to justify why an occupancy modeling approach was not used. One

reviewer provided suggestions on some ways to achieve that.

Please include metrics of model fit throughout, for example in Tables 1 and 2.

Looking forward to seeing the revision.

We appreciate the comments and suggestions from the reviewers and have added their names to our Acknowledgement section [Line 623-624]. We have provided point-by-point responses to the reviewer comments below.

With respect to our reasons for not using occupancy modelling, please see our response to Reviewer #2, Methods question 3 (page 9 below). We have also added metrics of model fit to the Results section to better link the table information to the text.

Journal Requirements:

1. Please ensure that your manuscript meets PLOS ONE's style requirements, including those

for file naming. 

We have reviewed the PLOS ONE style requirements and made the necessary changes to the file names. 

2. In your Methods section, please provide additional information regarding the permits you

obtained for the work. Please ensure you have included the full name of the authority that

approved the field site access and, if no permits were required, a brief statement explaining

why.

We have added “Work was completed under a CRD Regional Parks park use permit (#15/19) and signed consent to entry forms from all private landowners and T’Sou-ke Nation.” [Lines 198-200]

3. In your Data Availability statement, you have not specified where the minimal data set

underlying the results described in your manuscript can be found. PLOS defines a study's

minimal data set as the underlying data used to reach the conclusions drawn in the manuscript

and any additional data required to replicate the reported study findings in their entirety. All

PLOS journals require that the minimal data set be made fully available. For more information

about our data policy, please see http://journals.plos.org/plosone/s/data-availability.

We have now stored the minimal dataset on Dryad and included the citation “Klees van Bommel J, Sun CC, Ford AT, Todd M, Burton CA. Coexistence or conflict: black bear habitat use along an urban-wildland gradient; 2022 [cited 2022 Aug 18]. Database: Dryad [internet]. doi: 10.5061/dryad.nvx0k6dvf” [48] in our reference list. The dataset is currently listed as private for peer review and can be viewed using the following link: https://datadryad.org/stash/share/LX_zkxl9UuNjMggVpsaA2ukTn7j0646vs3KiBtF45IA

4. We note that you have stated that you will provide repository information for your data at

acceptance. Should your manuscript be accepted for publication, we will hold it until you

provide the relevant accession numbers or DOIs necessary to access your data. If you wish to

make changes to your Data Availability statement, please describe these changes in your cover

letter and we will update your Data Availability statement to reflect the information you

provide.

Please refer to the response to number 3 above. 

5. We note that Figure 1 in your submission contain [map/satellite] images which may be

copyrighted. All PLOS content is published under the Creative Commons Attribution License

(CC BY 4.0), which means that the manuscript, images, and Supporting Information files will

be freely available online, and any third party is permitted to access, download, copy,

distribute, and use these materials in any way, even commercially, with proper attribution. For

these reasons, we cannot publish previously copyrighted maps or satellite images created

using proprietary data, such as Google software (Google Maps, Street View, and Earth). For

more information, see our copyright guidelines: http://journals.plos.org/plosone/s/licensesand-copyright. 

The data used for land cover in Figure 1 is not copyrighted. We initially cited the dataset in our figure legend, but to avoid confusion have removed that line and now only include it in the supplemental table listing data sources as with all of the other environmental and anthropogenic variables used.

6. Please review your reference list to ensure that it is complete and correct. If you have cited

papers that have been retracted, please include the rationale for doing so in the manuscript

text, or remove these references and replace them with relevant current references. Any

changes to the reference list should be mentioned in the rebuttal letter that accompanies your

revised manuscript. If you need to cite a retracted article, indicate the article’s retracted status

in the References list and also include a citation and full reference for the retraction notice.

We have reviewed our reference list, fixed a couple of formatting issues, and found no retracted articles.

Reviewer #1: 

The authors analyze camera detection rates of black bears in and around a conflict hotspot. Urban-wildlife interfaces are expanding rapidly and understanding habitat use of key species in human environments is critical for their management. The manuscript is well-written, the objectives are clearly defined and addressed, and the findings are discussed

within a relevant context. I have few comments:

The authors acknowledge complexities with individual bear identification and discuss future

directions of research that include density estimates and individual movements. Is there

however any indication that multiple and enough individuals have been captured during your

study to be representative for the broader population. In particular because generalist species

typically display a great amount of behavioral plasticity, and the extent of the study area is

rather small (~80km2).

We have added the following to the Methods section [Line 178-180]: “Based on estimated female black bear home ranges on Vancouver Island (7.83 km2) [22], we expected the study area to include multiple overlapping home ranges, capturing a representative sample of the local bear population.” Additionally, in the Results we added “While individual bears are not reliably distinguishable from camera trap images, we did photograph multiple bears of different sizes at the same sites, sows with cubs of different sizes taken around the same date, and multiple photos taken close in time at different sites that were unlikely to be the same bear.” to Line 403-406. Therefore, we are confident we captured multiple individuals and that our inferences would extend beyond just a few bears to the broader population.

Related to above comment and a potential avenue for future research, would there be any

(expected) differences in habitat use between adult males, adult females, and females with

cubs? And how would this influence conflict or coexistence?

This is a good point. There is evidence from the literature suggesting that in general male black bears use areas of human development more than females and are thus more likely to be involved in human-carnivore conflict (Beckmann and Berger 2003, Merkle et al. 2013). However, one study found that while females used less developed areas in good natural food years, they selected higher development areas than males in poor natural food years (with and without cubs; Johnson et al. 2015). This could have an exacerbated negative effect on black bear populations if natural food sources are scarce and females are involved in more conflict which may lead to fatal mitigation methods. In fact, a population sink was identified in the Lake Tahoe Basin, Nevada, USA, because female bears in urban areas had higher age-specific mortality rates (Beckmann & Lackey 2008).

We have expanded on our suggestion for future research to look at individual variation within a population (and the implications for managing sex-specific conflicts) [Line 590-594]: “If bears are attracted to urban areas, determining the sexes of those individuals would be important as females of reproductive age have a greater impact on population demography. In general, male black bears have been found to use more developed areas and thus be involved in more conflict [12,16], however in poor natural food years, females may select areas of higher development than males [59].”

How did you deal with the fact that not detecting a bear at a camera trap location may be a

false absence? And consequently the zero’s in the zero inflated model may not all be true

absences, which may induce bias in inferences on habitat use?

We respond to this comment in detail below under Reviewer 2’s suggestion to use occupancy models to account for imperfect detection, please see our response under the section “Methods”, question 3 (page 9 below). 

L388-393: It is not clear to me on what base you make these inferences. You report, for

example, a negative association between bear habitat use and human density, but no effects of

trail density are reported. From looking at Fig3, it seems however that both variables have

similar coefficient estimates, standard errors (not overlapping zero) and confidence intervals

(overlapping zero). The same applies for the quadratic term of human density and elevation. I

would therefore think that only ‘Active Days’ and ‘EVI’ are informative predictors of bear

habitat use.

We thank the reviewer for catching this discrepancy. We interpret significant effects as those with a 95% confidence interval that does not overlap zero, so have reworded the Results to clarify that EVI is the only significant predictor variable and moved the model results for human density down with the other predictors [Line 422-424], and removed language stating human density was significant from the Abstract [Line 39-40] and Discussion [Line 500; 507; 517-519].

L414-418: Please be consistent in reporting p-values and could you include these in the

previous results section?

We thank the reviewer for catching this error and have included the p-values for the first set of models in the Results section as well [Line 420-431].

L416: maybe add ‘compared to spring’.

We have edited Line 449 to add “compared to spring”. 

L460-463 & L469-471: Is there any previous research in natural systems that shows seasonal

shifts in habitat use as a consequence of hyperphagia?

Yes, previous research has found that bears shift to areas of high natural food abundance before denning. We have added text to highlight that similarity on Line 509-511: “This is similar to observations of seasonal shifts in natural diets before hyperphagia, where bears select swamp habitat with an abundance of berries, grasses, and willows [50].”

Reviewer #2: 

In this paper the authors look at spatio-temporal variation in the distribution of black bears in

an area where conflict between black bears and humans is high. Additionally, they also look at

the correlation between black bear detections and the probability of black-bear conflict that

was estimated from previous research. In general, I think this is an interesting and well written

piece of research. I often found myself in agreement with the logical flow of the research, the

modeling choices made, and the interpretation of said models. Great work!

Perhaps my biggest concern is the secondary modeling approach, which treats the estimated

conflict probability as known, when in fact it is something that is estimated with error. In my

review below I have a number of hopefully helpful citations about this specific issue for the

authors.

I also suggest the authors look at a recent article of mine (Fidino et al. 2022), which shows

how integrated models could be use to combine human-wildlife conflict data with camera trap

data. I'm not on the hunt for a citation here, but since it is relevant to this piece of research I

am sharing in case you were interested in how to do this in the future.

Fidino, M., Lehrer, E. W., Kay, C. A., Yarmey, N. T., Murray, M. H., Fake, K., ... & Magle, S.

B. (2022). Integrated species distribution models reveal spatiotemporal patterns of human–

wildlife conflict. Ecological Applications, e2647.

Thank you for sharing this new paper! We agree this is a promising approach and we have added a citation to it in our recommendations for future research in Line 563-565: “Future research could further integrate reports of human-black bear conflict with camera trap surveys to more accurately predict and target where and when conflicts may happen, as in Fidino et al. [55].”

## Abstract

Line 46: The 'such as' breaks up this sentence in a weird way. I think you can remove it and

the examples become a little more clear.

We have edited Line 46 to removed “such as”. 

## Introduction

1. Well written introduction, I only have some minor comments on some of the wording /

sentence structure.

Line 64: Is sustainable the correct word here? I think it could be, but also it tends to have a bit

of a positive connotation to it, so it reads a little off to say sustainable risks of death. Maybe

'acceptable' could be used instead? Just a minor point, feel free to ignore if you disagree.

We can see how with a positive connotation, “sustainable” may be a jarring word to use in this context. However, the definition of coexistence we are referencing in this sentence and using in our paper was defined in the citation Carter & Linnell, 2016 [8], and we believe reflects the understanding that some conflict is unavoidable but should be mitigated to levels that sustain wildlife populations, human health, and livelihoods.

Line 69: This sentence is a little circular. Changes to avoid direct conflict can increase direct

conflict when risk of direct conflict is lower. I think it's the second use of direct conflict,

which makes it feel a little redundant. Could that bit right after the first (e.g.,) just get

removed? I think the point still comes across.

We have changed the sentence to remove the second instance of “direct conflict” and combined the examples in parentheses: “Behavioural changes to avoid direct conflict may actually increase indirect conflicts due to attempts by animals to access calorie-rich human food attractants (e.g., accessing garbage, fruit, or crops at night when interaction with people is less likely).” [Line 68-70]

Line 76-77: Is 'year round' needed when you already state they are available consistently?

"...as they are predictable, consistent, and spatially aggregated sources of..."

We have edited the sentence to remove “…available consistently (e.g. weekly), year-round, and are…” and replaced with: “…predictable, consistent, and…” [Line 76-77].

Line 82-84: Does it help here to remind the reader that the urban-wildlife interface is

increases, and therefore increased risk of ecological traps for such species?

We have edited the sentence from “These ecological traps function as population sinks which have negative impacts on long-term population persistence [14].” to “These ecological traps function as population sinks which have negative impacts on long-term population persistence [14], and may become more common as the urban-wildland interface increases.” [Line 84-85]

Line 88: ...to signal wildlife mortality risks…

The current line “Strategies include protecting anthropogenic food attractants or creating a hostile environment to signal the mortality risks for wildlife in human-dominated areas.” is meant to generalize human-carnivore conflict mitigation strategies that block or modify carnivore behaviour. In the latter case, techniques such as an electric fence (used as an example in the following sentence) provide an unpleasant or painful sensation to signal danger – this signal is directed at the wildlife in question and thus the phrasing “…to signal wildlife mortality risks…” would be unclear. To clarify our language, we have edited the word “for” to “to”, making the sentence read: “… to signal the mortality risks to wildlife…” [Line 89].

Line 98: ...and can alter their foraging patterns to…

We have edited “…the ability to quickly…” to the recommended “…can alter their…” [Line 99].

Line 103: ..., and can result in seasonal…

We have edited “but” to “and” as recommended [Line 104].

Line 103-105: Sentence starting with "If such attractants" reads a little weird. Maybe it's the

'attractants can attract animals' part or that the '...and food-conditioning' bit at the end feels like

a different thought tacked on.

We have changed the sentence to highlight the connection between attractant availability, reliance of bears on that food source, and behaviour change as a lead in to the next example: “If such attractants remain constant and available, they can draw animals from surrounding wild areas and create a reliance on anthropogenic food, which may change bear behaviour in the urban-wildland interface. In Nevada…” [Line 104-107].

Line 105 - 108: This sentence could be made more active (and therefore a little more

succinct). "In Nevada, urban black bears are less active each day (as they satisfied their caloric

needs faster), more nocturnal, and hibernate less."

We have replaced the sentence with the recommended text [Line 107-108].

Line 116-120: Conflict probability of what? Do you mean human-wildlife conflict probability

in general? Human-bear conflict in particular?

We have edited to clarify it is “human-black bear” conflict probability [Line 120].

Line 123-125: Great point, and the introduction sets a really strong foundation for the need to

study this.

## Methods

### Top-level thoughts

1. I would imagine that the distance between camera trapping locations would be species

specific to maintain spatial independence. For black bear, multiple camera traps can be

contained within the home range of a single bear, and therefore I would assume that spatial

independence is not met. This citation is also a pre-print, though it's been around for 12 years.

Maybe just soften the language here (e.g,. say to increase spatial independence rather than

maintain spatial independence).

We changed “maintain” to “increase” as suggested [Line 191], and we corrected the error in our citation (it is published in a journal, not a pre-print).

Kays, R., Tilak, S., Kranstauber, B., Jansen, P., Carbone, C., Rowcliffe, M., Fountain, T., Eggert, J., He, Z., 2011. Camera traps as sensor networks for monitoring animal communities. International Journal of Research and Reviews in Wireless Sensor Networks 1, 19-29.

While we agree that individual bears could be detected at multiple camera traps, we are suggesting that detections are statistically independent as measures of habitat use (as evidenced by the lack of spatial autocorrelation in detections). If individual identities were known, a random effect could be used to account for between-individual variation, but that is not possible with our data.

2. Was the sampling effort term included as an offset to the model or did you just include it as

a covariate? If the latter was done, I would encourage adding a log offset term instead, as that

is a more standard approach when there is variation in sampling effort. If I had to guess, you

used a log-offset, so just be specific about that here in the methods.

We have updated the models to include Active Days as an offset instead of as a covariate and subsequently removed Active Days from Figure 3. The model’s results remain the same, with the conflict model still the top model and parameters estimates nearly identical (i.e. only changing by tenths or hundredths of a decimal place). The Methods and Results have been edited to reflect these changes, with text to clarify our modelling approach [Line 240-242]: “All models also included the number of active camera days as an offset to account for variable sampling effort (i.e. not all cameras were active for all study days).” and updated estimates in the Results [Line 416-431], Table 1, and Table S3.

3. I could see some readers get their hackles up about not using an occupancy model here. I

leave it to the authors, but it may help to just get ahead of it here (by using a zero-inflated

model you may be estimating habitat use conditional on presence anyways, depending on the

class of zero-inflated model you fit). Likewise, occupancy models are literally just zeroinflated

logistic regression, so your approach has substantial overlap. For example, the old

MacKenzie et al. occupancy modeling book talks about these similarities on about page 135

(i.e., the use of the zero-inflated binomial to model occupancy).

MacKenzie, D. I., Nichols, J. D., Royle, J. A., Pollock, K. H., Bailey, L. L., & Hines, J. E.

(2017). Occupancy estimation and modeling: inferring patterns and dynamics of species

occurrence. Elsevier.

We appreciate that the use of occupancy models is very popular in the camera trap literature, however we feel that they are often applied uncritically and their reliability is an area of active research. We chose to directly model detection rates as measure of site habitat use, since occupancy models have an unsupported assumption of site closure and treat heterogeneity in detections within (arbitrarily defined) sampling periods as observation error, whereas we think it more reflects the signal of interest. Furthermore, some studies have suggested that estimates from occupancy models may not be reliable, particularly for species with relatively large home ranges relative to the spacing of sampling points (e.g., Neilson et al. 2018). We have added some text to clarify our thinking in manuscript on Line 233-235: “We chose to directly model detection rates instead of commonly proposed occupancy models as estimates from the latter may not be reliable for species with relatively large home ranges compared to the spacing of sampling points [33].”

4. There are a few different kinds of zero-inflated glmms, can you be a little more specific?

For example, what is the error distribution that was used (negative binomial, Poisson, etc.). Is

this a two part model and therefore the conditional model cannot include zeroes or is it a

mixture model and therefore the conditional models can include zero? A little more

explanation here would be helpful. note: I see now that the distributional information is shared

much lower, around line 297. I'd move that little bit of info upwards so it's not separated from

when you introduce the modeling framework.

We moved up the text explaining the negative binomial distribution earlier, as suggested, from Line 294-298 to Line 242-245: “Models were run with the ‘glmmTMB’ package [35] in Program R [36]. We used the negative binomial distribution “Nbinom2” which treats the variance quadratically because all candidate models had a lower AICc compared to the “Nbinom1” distribution (where variance was treated linearly) [35].”

Further, the package glmmTMB runs the zero-inflated model as a mixture model, so the conditional models can include zero. For more information see:

Brooks, M. E., Kristensen, K., van Benthem, K. J., Magnusson, A., Berg, C. W., Nielsen, A., Skaug, H. J., Machler, M., & Bolker, B. M. (2017). glmmTMB balances speed and flexibility among packages for Zero-inflated Generalized Linear Mixed Modeling. The R Journal, 9(2), 378-400. https://doi.org/10.32614/RJ-2017-066

5. Thank you for using a set of candidate models to assess your different hypotheses. Solid

approach. Given the two set hypotheses brought up in the introduction, in may help to add a

little more connection between those hypotheses and the set of candidate models given that

there are two hypotheses but four candidate models. You should also fit a fifth null model as

well (just the active days term plus the site random effect). Given the results I suspect the null

will provide the worst fit (highest delta AIC), but it's nice to demonstrate this.

We have added some clarifying language to the Methods section outlining our habitat use models to connect the four candidate models with the overarching two hypotheses to Line 284-289: “We expected if our overall “conflict hypothesis” was supported, the best fit model would show selection for human-dominated areas with high values for variables associated with high conflict probability; selection for wild areas with high values for low conflict probability variables, would support our “coexistence hypothesis”. We tested four sets of candidate models to see which types of predictor variables best explained bear habitat use.”

We also edited the table caption for Table 1 from “Model sets are grouped by hypothesis…” to “Models are grouped by candidate set…” [Line 306-307].

We did run a null model and have now included mention of that in our Methods [Line 299] as well as Table 1. The reviewer is correct in that the null did have the highest delta AIC (24.1 which is more than 10 dAIC higher than the next model, see Table 1). 

6. Based on the introduction (lines 133-134) I thought that modeled conflict probabilities

would be incorporated into your glmms, but instead it looks like counts of conflict are

included instead. I suspect other readers will also be confused about this. What makes this

more confusing, is that there is a second batch of models done that uses the estimated conflict probabilities.

We used counts of conflict reported within the year of camera trapping in the first set of models to incorporate local observations from the same temporal scale as the habitat use dependent variable. For the second set of models, we wanted to test how well the predictions from a regional model built using seven years of previous conflict reports matched the local habitat use results from the first set of models. To clarify, we replaced the word “also” with “subsequently” in Line 137: “To assess the relationship between habitat use and conflict, we subsequently modeled detections using previously estimated seasonal conflict probabilities from the same region [18].” 

 We have also added some text in the Methods section for the second set of models to state our intention to test the regional conflict probabilities against local habitat use [Line 326-327]: “…, and thus test predictions from a regional model of conflict at a local scale…”.

7. Why conflicts over the year instead of conflicts per month?

We used the number of reported conflicts within a 500 m buffer of a camera site within the study year as one of our additional local-scale test variables. We decided to use conflicts per year instead of per month because there were not enough reports per month at most of our sites to include.

8. Using the output from one model as a predictor in another is okay, but the uncertainty of

those estimates should also be propagated into the secondary model. From my reading of this

secondary model set, I'm guessing that these spatially explicit probabilities are treated as

known (i.e., measured without error). If this is the case, using such predictions in secondary

analysis leads to anticonservative tests because this error is excluded from further tests (i.e.,

estimates are too precise). Some papers about this topic that the authors may find useful

include:

Hadfield, J. D., Wilson, A. J., Garant, D., Sheldon, B. C., & Kruuk, L. E. (2010). The

misuse of BLUP in ecology and evolution. The American Naturalist, 175(1), 116-125.

Houslay, T. M., & Wilson, A. J. (2017). Avoiding the misuse of BLUP in behavioural

ecology. Behavioral Ecology, 28(4), 948-952.

Link, W. A. (1999). Modeling pattern in collections of parameters. The Journal of

wildlife management, 1017-1027.

The Houslay & Wilson paper is open access, so that is where I'd start. I've personally found

this easiest to account for in a Bayesian framework (e.g., if you have the mean and SE of each

prediction you can set a prior for each data point to propagate that uncertainty), but there are

likely ways to deal with this in a frequentist framework as well (e.g., bootstrapping, but

resampling the predicted covariate instead of the response variable).

Thank you for this reflection and the useful references. We acknowledge that we did not propagate the uncertainty from the previously published conflict models into this secondary modelling in this study, and that this is an important consideration. However, this would require an overhaul of our modelling approach, and the addition of further details of the previous models, that is beyond the scope of our intentions for this manuscript. We anticipate that local conflict managers are more likely to use maps of the mean estimates of predicted conflict probabilities in land use planning, i.e. to delineate relatively higher vs lower predicted probabilities. Our intention was to see if these mean predictions from the previously published regional models corresponded to our estimates of local black bear habitat use, in the context of comparing competing model-based hypotheses, rather than to estimate precise coefficients from this second stage of modelling. Thus, we felt it was reasonable to use the mean predictions as “certain” in the way managers might for planning purposes. However, we have added text to clarify that we did not propagate uncertainty and suggested that future models could better integrate conflict and habitat use (Line 561-565): “We do note that our second set of models comparing habitat use to seasonal conflict probability did not propagate the uncertainty from the models used to estimate the conflict probabilities. Future research could further integrate reports of human-black bear conflict with camera trap surveys to more accurately predict and target where and when conflicts may happen, as in Fidino et al. [55].”

9. What is a reported conflict in these data? Do they vary in severity?

We defined conflicts as any interaction that has a negative impact on either black bears or humans. Common conflicts with bears included accessing garbage and other anthropogenic food sources, property damage, and livestock predation. These conflicts are further explained in Klees van Bommel et al. 2020, but we noted “garbage attractants to include fruit, compost, and other rural attractants” along with the citation on Line 123-124. Livestock predation may be considered greater severity, but we did not attempt to rank conflicts. 

## Results

### Top-level thoughts

1. Failing to detect an effect does not mean that there was no effect (e.g., line 391 - 392). I'd

just reword to "We failed to detect an effect of road density" so that you avoid confirming the

null (which these tests do not do). Other than that, great breakdown of the results.

We edited as recommended [Line 422].

## Discussion

### Top-level thoughts

1. Given the conflicting hypotheses, do the authors feel that one hypothesis was supported

more than the other?

We believe that further research is required to strengthen support for one hypothesis over the other. We recommended extending the length of camera trap sampling and the addition of finer scale data on anthropogenic and natural bear foods – particularly salmon and berries – to determine the role that natural food availability plays in bear conflict behaviours [Lines 594-598]. However, given the conflicting support among hypotheses in our study, it is likely, that new hypotheses that explicit address behavioural differences across seasons, sex, and ages should also be tested. We provide such recommendations for future research, as well as the recommendation to test if urban areas are sinks affecting the broader black bear community and if conflicts result from widespread behaviours or are limited to certain “conflict bears”. 

2. Any caveats worth bringing up here? For example, there was the assumption that EVI

indicates forage availability. Is it possible for there to be human-bear conflicts that go

unreported and so the conflicts / year metric used may have some error?

Yes, to clarify our assumption the EVI indicated forage availability and crop ripeness, we edited Line 550-553 to read: “If our assumption that higher EVI indicates forage availability and crop ripeness is correct, as has been shown in other studies [12,38,39], Sooke bears may be selecting for…”

We also agree it is important to emphasize that conflict is reported with uncertainty and have included text in Line 565-570: “Conflict reports themselves are a sample of all the conflict that occurs, and thus contain error as some conflict goes unreported. However, community demographics have been found to have limited influence on the chance of reporting conflicts, and conflicts relating to safety or property damage (which encompass many human-black bear conflicts) are more likely to be reported [18]. We therefore assumed that sampling of conflicts across Sooke was not systematically biased.”

### Line by line comments

Line 530: You used ecological trap earlier.

We edited “sink” to “trap” [Line 587].

## Tables & figures

### Top-level thoughts

1. The axis text on many of the figures is a very light gray, I'd suggest replacing with black to

make it easier to read.

The axis text is in black, but in some cases the axis label is in a larger font and bolded, so the axis text may appear lighter in comparison. We have increased the font size for the axis text on figures 3-5 to make them clearer.

2. You could increase the line width for the 95% CI's on figure 3&4, plus the mean estimate

on figure 5.

We have doubled the width of the 95% confidence interval lines for Figures 3 and 4, as well as for the mean estimates in Figure 5. 

Additional Edits

We have updated the author affiliation for co-author Melissa Todd on Line 15.

Line 132 citation changed from “[British Columbia Conservation Officer Service, unpublished data]” to “[18]” as the results were previously published in Klees van Bommel et al. 2020.

We edited Line 158-164 to clarify the findings of black bear hunting data, from “Vancouver Island, British Columbia, Canada, is home to black bears living at high densities near urban areas. While recent bear population estimates for Vancouver Island are not available, high bear abundance is indicated by some of the highest hunter harvesting rates in the province, increasing from 300 to 700 bears per year since the 1980s with no change in hunter success [23]. The municipality of Sooke, on the southern tip of Vancouver Island, British Columbia, Canada,...” to “Vancouver Island, BC, Canada, is home to black bears living at high densities near urban areas. Recent bear population estimates for Vancouver Island are not available, however high bear abundance is indicated by some of the highest average annual harvest densities across BC during the past ten years (up to 25 bears/100 km2) [23]. The municipality of Sooke, on the southern tip of Vancouver Island,…”

We have further edited our explanations of EVI in a couple locations for improved clarity. Firstly, in Line 262-263 we specified that EVI is a proxy for fruit abundance as greenness peaks at the same time: “EVI has been used as a proxy for fruit abundance (grapes, Vitis spp.) in rural areas as ripeness peaks at the same time as greenness [39].” Secondly, we added “...as a proxy for food and cover.” to Line 293.

We also clarified that we didn’t include counts of conflict in our “conflict” habitat use model like we did in our “anthropogenic” model because we wanted to test if the same predictors from Klees van Bommel et al. 2020 were also the best fit for these data in Line 290-292: “...using the same predictors as Klees van Bommel et al. [18] used to model regional-scale on human-black bear conflict in the same study area, but applied to the local camera scale…”

To clarify why spring is not shown in Fig 4, we added text to the Methods in Line 337-338: “Seasons were modelled as a factor, with spring used as the intercept.”

We added a note that the scale at which we extracted variables may have contributed to many variables having an insignificant impact on habitat use [Line 524-525]: “The lack of a similar effect in our study may have resulted from the 150 m buffer size we extracted our variables at, or…”

Finally, some of these edits were suggested by Garth Mowat who provided comments through our co-author Melissa Todd. We have thus included

Response to Map Copyright Question

1. Please note that PLOS ONE is unable to publish previously copyrighted maps or satellite images, or images created using proprietary data. For these reasons, we cannot publish images generated by software which copyrights their output (such as Google Maps, Street View, and Earth). In order to use these images in your submission, we require explicit permission from the copyright owner to publish the figures under the CC BY 4.0 license.

At this time, please kindly clarify the following regarding Figure 1:

a) Where did the authors obtain the maps, satellite images, basemaps, shapefiles, map data, etc. in Figure 1?

The map was created by author Joanna Klees van Bommel using ArcGIS Pro. The camera locations are data collected by author Joanna Klees van Bommel while setting the camera traps for the duration of the project. The land cover dataset which serves as a basemap and the Parks shapefile were obtained from CRD Regional Parks. The T’Sou-ke Nation Land shapefile was adapted from the “Aboriginal Lands of Canada Legislative Boundaries” dataset by the Government of Canada.

b) Please state whether the map/satellite images have been previously copyrighted to your knowledge.

The map was created by author Joanna Klees van Bommel using ArcGIS Pro. According to their website: “You do not need to obtain permission from Esri to include static maps, whether screen capture or printed, in academic publications, for personal use, or in most use cases that do not involve direct resale or commercial monetization of the map.” https://resources.esri.ca/education-and-research/how-to-cite-arcgis-maps-and-data

The camera locations are data collected by author Joanna Klees van Bommel while setting the camera traps for the duration of the project and are not copyrighted.

The land cover dataset and Parks shapefile were obtained from CRD Regional Parks and are not copyrighted. 

The T’Sou-ke Nation Land shapefile was adapted from the “Aboriginal Lands of Canada Legislative Boundaries” dataset licenced under an Open Government Licence – Canada which allows for uses including to “Copy, modify, publish, translate, adapt, distribute or otherwise use the Information in any medium, mode or format for any lawful purpose.” https://open.canada.ca/en/open-government-licence-canada

c) If any of the map/satellite images have been previously copyrighted, we require specific consent from the copyright holder to publish these images in PLOS ONE, under the CC BY 4.0 license. To seek permission from the copyright owner to publish your map figures under the specific Creative Commons Attribution License (CCAL), CC BY 4.0, please contact them with the following text and PLOS ONE Request for Permission form (http://journals.plos.org/plosone/s/file?id=7c09/content-permission-form.pdf):

“I request permission for the open-access journal PLOS ONE to publish XXX under the Creative Commons Attribution License (CCAL) CC BY 4.0 (http://creativecommons.org/licenses/by/4.0/). Please be aware that this license allows unrestricted use and distribution, even commercially, by third parties. Please reply and provide explicit written permission to publish XXX under a CC BY license.”

Please upload the granted permission to the manuscript as an other file. In the figure caption of the copyrighted figure, please include the following text: “Republished from [ref] under a CC BY license, with permission from [name of publisher], original copyright [original copyright year].”

Not applicable.

---

## [Editor Report · Decision Letter 1]

7 Oct 2022

Coexistence or conflict: black bear habitat use along an urban-wildland gradient

PONE-D-22-12898R1

Dear Dr. Klees van Bommel,

We’re pleased to inform you that your manuscript has been judged scientifically suitable for publication and will be formally accepted for publication once it meets all outstanding technical requirements.

Kind regards,

Bogdan Cristescu

Academic Editor

PLOS ONE

Additional Editor Comments:

The revisions helped improve clarity and strengthened the manuscript. Congratulations on your paper.

---

## [Editor Report · Acceptance letter]

18 Nov 2022

PONE-D-22-12898R1 

Coexistence or conflict: black bear habitat use along an urban-wildland gradient 

Dear Dr. Klees van Bommel:

I'm pleased to inform you that your manuscript has been deemed suitable for publication in PLOS ONE. Congratulations! Your manuscript is now with our production department. 

Kind regards, 

on behalf of

Dr. Bogdan Cristescu 

Academic Editor

PLOS ONE